# A mixed-method analysis of provider adherence to integrated antenatal care guideline in BEmONC and Non BEmONC primary health center: An Indonesian case

**Suryani Yuliyanti**[1]*, **Adi Utarini**[2]☯, **Laksono Trisnantoro**[2]☯

1 Public Health Department, Faculty of Medicine, Universitas Islam Sultan Agung, Semarang, Indonesia,
2 Department of Health Policy and Management, Faculty of Medicine, Public Health and Nursing, Universitas Gadjah Mada, Special Region of Yogyakarta, Indonesia

☯ These authors contributed equally to this work.
* suryaniyuliyanti@unissula.ac.id

**Data Availability Statement:** All relevant data are within the manuscript and its Supporting information files.

## Abstract

Provider adherence to the integrated antenatal care (ANC) procedure is an important indicator of high-quality ANC. The procedure is intended to avoid missed opportunities to detect the risk of abnormalities in pregnancy. This study aims to assess the provider's adherence to integrated ANC in Basic Emergency Obstetric and Newborn Care (BEmONC) and non-BEmONC Primary Health Center (PHC). This study employed an explanatory sequential mixed-method design. The quantitative phase reviewed 149 medical records of pregnant women in the four PHCs in Semarang from January until February 2020. The findings were used to describe the provider's adherence to the integrated ANC and lead to the contributing factors which should be explored in the qualitative phase. The study involved four in-depth interviews with midwife coordinators in four PHCs. The Mann-Whitney and Chi-square test was employed to analyse the quantitative data, while the thematic analysis was undertaken on the qualitative data. The provider's adherence to the guideline did not differ between BEmONC and non-BEmONC PHC. The general physical examination of the patients (18.81%) and dentist visits (84.6%) were not done in either BEmONC or non-BEmONC PHC. Incomplete laboratory tests were haemoglobin (28.2%) and urine protein (38.9%). The barriers to adherence to the integrated ANC guideline were related to an imbalance of resources, role and responsibility issues among doctors and midwives, and policy issues. This study found low BEmONC nor non-BEmONC PHC adherence to the integrated ANC guideline. A Periodic evaluation of the implementation of integrated ANC to ensure its high-quality implementation in Indonesia is needed.

## Introduction

Antenatal care (ANC) has been associated with reduced Maternal Mortality Rate (MMR) through early detection of abnormalities in pregnancy followed by proper interventions [1–3].

**Funding:** SY. Universitas Islam Sultan Agung. 244/
B.1/SA-LPPM/VIII/2020 The funders had no role in
study design, data collection and analysis, decision
to publish, or preparation of the manuscript.

**Competing interests:** The authors have declared
that no competing interests exist.

Various disorders during pregnancy can be detected early through the high-quality integrated
ANC to avoid missed opportunities, provide timely treatment, and adequate education-related
pregnancy and postnatal care to improve the outcome [3, 4]. Since 2016, the World Health
Organization (WHO) has recommended at least eight ANC visits during pregnancy and using
a core set of ANC services for safe motherhood [3]. However, most developing countries,
including Indonesia, still enforce previous WHO recommendations requiring at least four
ANC visits during pregnancy during the last three preceding years [5]. Earlier studies in devel-
oping countries reported the poor quality of ANC content and timing of the first ANC. Studies
in Pakistan [6], South Africa [7], Nepal [8], Bangladesh [9], Ethiopia [5], and Uganda [10]
reported that 17.1% to 30% of mothers received all eight selected elements of ANC services,
low level of WHO minimum recommended ANC [11]. The study suggested the importance of
identifying ANC fulfilment determinant factors, including provider adherence to the ANC
guideline [12, 13].

In Indonesia, maternal death is reported in the four most populated provinces [14] (i.e.,
Bali, West Java, East Java, and Central Java) with the highest coverage of at least four times
ANC visit (94.6%) [15]. Despite the high range of routine ANC visits, Semarang, Central Java
province's capital's Maternal Mortality Rate (MMR), ranks fourth [16]. This aligns with a pre-
vious study conducted in Aceh, Indonesia, which indicates that the frequency of ANC visits
more than four times did not predict a good outcome in pregnancy. Instead, it is more likely
to be associated with the quality of ANC [17]. Inadequate and incomplete ANC examination
increases the risk of maternal death up to 7.86 times. Moreover, mothers who have never had
an ANC have a 3.5 times greater risk of experiencing complications than those who have had
at least four times or more ANC examinations [18]. Most ANC programs established in low
and middle-income countries are largely underachieved [5]. Research in Uganda detected that
pregnant women on the mainland and women who visited the doctors received more com-
pleted ANC procedures and performed ANC visits more often [10]. Based on the national
Basic Health Research data in 2018, 74% of women in Indonesia had four or more ANC visits.
However, only 69% of pregnant women receive the full content of integrated ANC during
their pregnancy. In Central Java, Indonesia, 82% of women had four or more ANC visits, and
74% received completely integrated ANC services [16]. Furthermore, among those with four
times or more ANC visits, only 4% received a complete integrated ANC [8].

Numerous studies have been conducted to identify the factors that contribute to maternal
mortality rates, and one of these factors was the quality of antenatal care (ANC) provided [19].
Although some studies have reported on the quality of ANC services, they have mainly focused
on fulfilling the requirement of four visits during pregnancy [15, 20–22]. Only a few studies
have evaluated adherence to national antenatal care guidelines [19, 23, 24]. However, it is
essential to note that the quality of ANC can be assessed by ensuring that all examination pro-
cedures are adequately fulfilled [25, 26].

Integrated ANC is a comprehensive and integrative service to fulfil pregnant women's
rights and promote a healthy pregnancy, safe delivery, and healthy babies. All pregnant
women must obtain the ANC as the minimum service standard set by the Indonesian govern-
ment [27, 28]. The mandatory examinations in the ANC included physical examination, blood
test, education, and case management (see Table 1) [29]. In addition, it requires collaborative
care among general practitioners, dentists, and nutritionists in the Primary Health Center
(PHC/Puskesmas) [28, 29]. Although the government had enacted the mandatory implemen-
tation of integrated ANC in 2014 [28], only 37–70% of pregnant women in Indonesia received
integrated ANC [15, 30].

In Indonesia, there are two types of Primary Health Centers (Puskesmas): those with Basic
Emergency Obstetric and Newborn Services (BEmONC/ Pelayanan Obstetri Neonatal

**Table 1. Guidelines for a recommended package of services to be provided by antenatal care (ANC) facilities.**

| No | Examination | 1st Trimester | 2nd Trimester | 3rd Trimester | 3rd Trimester |
|---|---|---|---|---|---|
| 1 | General condition | ✓ | ✓ | ✓ | ✓ |
| 2 | Temperature | ✓ | ✓ | ✓ | ✓ |
| 3 | Blood pressure | ✓ | ✓ | ✓ | ✓ |
| 4 | Body mass | ✓ | ✓ | ✓ | ✓ |
| 5 | Mid upper arm circumference | ✓ | | | |
| 6 | Height of fundus uteri | - | ✓ | ✓ | ✓ |
| 7 | Fetal presentation | - | ✓ | ✓ | ✓ |
| 8 | Fetal heart rate | - | ✓ | ✓ | ✓ |
| 9 | Hemoglobin test | ✓ | * | ✓ | * |
| 10 | Blood type | ✓ | - | - | - |
| 11 | Urine protein level | * | ✓* | ✓* | * |
| 12 | Blood glucose/reduction | * | ✓* | ✓* | * |
| 13 | Hepatitis | ✓* | - | - | - |
| 14 | Malaria test | * | * | * | * |
| 15 | Acid-fast bacilli test (AFB) | * | * | * | * |
| 16 | Syphilis test | * | - | - | - |
| 17 | HIV serology | ✓* | - | - | - |
| 18 | USG | * | * | * | * |
| Interdisciplinary examination and education | | | | | |
| 19 | Midwives | ✓* | ✓* | ✓* | ✓* |
| 20 | General practitioner | ✓* | ✓* | ✓* | ✓* |
| 21 | Nutritionist | ✓* | * | * | * |
| 22 | Dentist | ✓* | * | * | * |

✓ a recommended minimum package of services to be provided by antenatal care (ANC) facilities. (national policy and/or Ministry of Health (MOH) guidelines)

* a recommended minimum package of services to be provided by antenatal care (ANC) facilities based on the indication (national policy and/or Ministry of Health (MOH) guidelines)

✓* recommended package of services to be provided by antenatal care (ANC) facilities. (local policy guidelines/municipal health officer)

Emergensi Dasar (PONED)) and those without. A BEmONC Primary Health Center provides additional inpatient and emergency services for mothers and newborns 24 hours a day, seven days a week (24/7) [3, 31, 32]. Both types of PHC should provide a high-quality integrated ANC and adhere to the national minimum healthcare standards despite facility and staffing differences [24, 32]. Previous descriptive studies showed a difference in the achievement of minimum health care standards between the two types of PHC, with more women having antenatal care at least four (4th ANC) visits in BEmONC than non-BEmONC PHC [30]. However, limited studies have compared the performance of BEmONC and non-BEmONC PHC in delivering integrated ANC. This study aims to test the hypothesis that, compared with non-BEmONC PHC, BEmONC PHC provides better integrated antenatal care and explores factors associated with provider adherence.

## Materials and methods

### Study setting

A Primary Health Center (PHC/Puskesmas) is a unit of the district/city Health Service in Indonesia responsible for Health Development in its working area [33, 34]. One of the mandatory programs of PHC is antenatal examination, which must have 100% coverage. The quality

of care for pregnant women is measured by fulfilling all antenatal care procedures, not just limited to the number of visits. There are three types of PHCs in Indonesia: outpatient-only health centers, inpatient health centers, and auxiliary health centers. An inpatient health center is a strategically located health center accessible by public transportation and equipped with standard inpatient facilities and infrastructure. A qualified BEmONC PHC offers midwifery and basic health services, emergency services and neonatal complications at a basic level, 24 hours a day and seven days a week. Referral services for pregnant women involve collaboration between BEmONC PHC and hospitals providing Comprehensive Maternal and Neonatal Emergency care (CEmONC/ Pelayanan Obstetri Neonatal Emergensi Komprehensif (PONEK)) to improve the quality of services in an integrated and comprehensive manner and reduce maternal mortality rates. The scope of a BEmONC PHC includes service management, a referral system, and community mobilisation through health partners in its working area, including the infrastructure needed to become a BEmONC PHC. BEmONC PHCs have better resources and facilities compared to non-BEmONC PHCs (inpatient health centers and non-inpatient health centers) [35]. Therefore, it is expected that BEmONC PHCs will provide more comprehensive ANC services and comply with all ANC service procedures better than non-BEmONC PHCs. However, government regulations mandate that both health centers carry out all integrated ANC inspection procedures.

The maternal mortality rate in Indonesia has been decreasing since 1991, going down from 390 to 228 in 2011. Although it increased to 359 in 2015, it decreased again to reach its lowest figure of 189 in 2020 [36]. Similarly, in Semarang City, the maternal mortality rate has consistently decreased from 128 in 2015 to 71.35 in 2020, successfully achieving the national target. However, there was an alarming spike in 2021, with the rate increasing to 95.30, but fortunately, the number of maternal deaths significantly decreased to 67.25 in 2022 [34, 37]. Although cases of maternal mortality have decreased in Semarang City, this issue still requires special attention since the city of Semarang is the capital of Central Java Province, which sets an example for the surrounding areas and the entire province. To reduce maternal mortality, the Semarang City Government is implementing one program called SAN PIISAN (SAyaNgi and damPingi Ibu & Anak SemarANg City / caring and assisting mothers and Children in Semarang City), which is a comprehensive health program that assists individuals during the first 1000 days of life, starting from teenagers, prospective brides, pregnant women, giving birth, post-partum, infants, and toddlers under three months of age. The service-oriented paradigm prioritises services based on needs and involves various stakeholders by moving together. Additionally, the SAN PIISAN Innovation Program helps identify causes of maternal and infant mortality, such as pregnant women who are too young (<20 years), too old (>35 years), have more than four children, have had less than four previous pregnancies, and three delays in referrals. This innovation has been implemented since 2017 and continues to this day with the goal of reducing maternal and infant mortality rates [34].

Semarang, the capital of Central Java Province, Indonesia, has a population of 1,668,578 people, consisting of 825,964 male residents and 842,615 female residents. There are 37 accredited PHCs and 37 auxiliary PHCs. Eleven out of 37 PHCs have inpatient facilities, including six BEmONC PHCs. The local government's efforts to reduce MMR in the preceding four years include free ANC service in PHC for women with Semarang citizenship. The government also requires every Comprehensive Emergency Obstetric and Newborn Care (CEmONC) hospital to regularly train their network of primary health center providers on maternal referral and emergency management based on the problem in referral cases. Besides, the local government recruits midwives annually to survey and assist pregnant women during their pregnancy [34, 38]. Furthermore, the local government has enacted the modified integrated ANC as a mandatory examination for every pregnant woman who visits the PHC. It

involved a physician, dentist, midwife, and nutritionist who collaboratively examined body weight, height, blood pressure, upper arm circumference, uterine fundal height, fetal presentation, and fetal heart rate to determine the nutritional status of women and babies. They also handled pregnancy management cases, tetanus immunisation, blood supplement tablet prescription, laboratory tests, and Interviews/counseling [27, 34, 37].

## Study design

This study employed an explanatory sequential mixed method design. The design was explanatory and sequential in that the quantitative phase preceded the qualitative phase. Both quantitative and qualitative data are equally presented to provide a comprehensive overview of integrated antenatal care implementation, including the factors that influence it at both types of primary healthcare facilities [39]. The integration of both quantitative and qualitative data was presented in the discussion section [40]. This study was part of the participatory action research on the implementation of interprofessional collaboration in caring for pregnant women with heart diseases [41] registered at the ISRCTN registry (https://www.isrctn.com/ISRCTN82300061) [42]. The four PHC facilities (Table 7) linked to the tertiary health care facilities participating in the action research to implement an integrated care pathway on pregnant women with cardiac disease [41] were selected. We have chosen the PHCs that collaborate with the hospital where the principal researcher works. This approach leads to an extended engagement, which significantly improves the intrinsic qualities of the qualitative data collected [43]. The protocol of this study was approved by the Medical/Health Research Bioethics Commission of the Universitas Islam Sultan Agung (approval number 817/XII/2019/Komisi Bioetik). All methods followed the relevant guidelines and regulations under the Bioethics Commission of the Universitas Islam Sultan Agung research policy. Written informed consent was obtained from all health provider participants, and oral informed consent was obtained from pregnant women selected for observation at their antenatal care consultations at the four primary health centers (PHCs) in Semarang. Participants were informed of their right to withdraw from the study at any point and that the information obtained was confidential. For medical record review, patients' names or PHC registration numbers were not used to ensure confidentiality and access to participants' information was not given to the author. The antenatal care adherence was taken from anonymised e-medical record recapitulation in the primary health center. Hence, the authors can only access antenatal care examination procedures without knowing the patient's identity.

## Quantitative phase

In the quantitative phase, the data on the socio-demographic characteristics of pregnant women and a package of services provided by antenatal care (ANC) of four PHCs in Semarang, Indonesia, were obtained from medical records from November 2018 to August 2019 from January to February 2020 by SY and the research assistant (RA). The minimum sample size was estimated using a formula for the two-population hypothesis testing [44]. Due to limited studies comparing the two types of PHC regarding integrated ANC performance, we referred to a study conducted by Afrizal et al. [19], who evaluated the adequacy of the cohort ANC registry documentation in South Tangerang, Indonesia. To assess the differences in the level of provider adherence to the integrated ANC, we used α = 5, 1-β = 90, P1 (percentage of adequate cohort ANC register in PHC without inpatient care unit) = 0,207 and P2 (percentage of adequate cohort ANC register in PHC with inpatient care unit) = 0,05. The minimum sample calculated from the formula was 94 medical records of pregnant women with at least four ANC visits.

Provider adherence to integrated ANC guidelines was evaluated using modified national integrated antenatal care guidelines. According to the regional regulation, an integrated antenatal care guideline should include an interdisciplinary examination and education. The national standard for integrated antenatal care was modified according to the local policy of the regional regulation of the study setting [35]. The modified guideline consisted of a package of services to be provided by antenatal care (ANC) facilities: 16 recommended minimum packages of services enacted by the Indonesian Ministry of Health and six recommended packages enacted by local policy guidelines/municipal health officer. The modified guideline consisted of 45 examination procedures, i.e., 13 items performed in the first trimester, 11 for the second trimester, and 21 for the third trimester (Table 2) [35]. Each item was assigned a score of 1 for adherence and 0 for non-adherence. Adherence was defined as complete adherence to the guidelines, with a total score of 45. A score of less than 45 was categorised as non-adherence.

The patient's socio-demographic information on the medical record, including age, education level, obstetrical status (gravida, para, abortus), and PHC status, were descriptively analysed. Adherence to guidelines was assessed based on the fulfilment of items in the modified national integrated ANC guideline checklist. Data were expressed as mean (standard deviation; SD), minimum-maximum, number (n), and percentage (%) where appropriate; $p < 0.05$ was considered statistically significant. Statistical analysis was employed using IBM SPSS Statistics for Windows, Version 22.0 software (IBM Corp., Armonk, NY, USA). Fisher's exact test (for 2×2 tables where some of the expected counts are less than 10), Likelihood Ratio (for bigger tables where 20% or more of the expected count is less than 5), and Pearson Chi-Square test were used to analyse the association of categorical variables. The provider adherence score was analysed using the Mann-Whitney U test (Kolmogorov-Smirnov test p = 0,000). The selection of hypothesis tests is based on a variable scale, categorised into 2 to 3 groups with commonly used demographic characteristics [40, 45].

## Qualitative phase

During the qualitative stage of the study, the first author (SY) conducted four in-depth interviews with midwife coordinators to explore the barriers that prevent providers from adhering to the integrated ANC guideline. These coordinators were purposively selected based on their knowledge, experience, and authority in their respective workplaces. Pre-arranged questions for in-depth interviews were modified based on the findings of the quantitative phase, namely: What are the obstacles and the promoting factors to the provider adherence to the integrated ANC guideline?

The data gathered from in-depth interviews with midwife coordinators were cross-checked with other sources, including reviews of national and regional antenatal care policies, observations of antenatal care facilities, interviews with three conveniently selected pregnant women who made antenatal care visits, and in-depth interviews with laboratory analysts and doctors, thereby ensuring its accuracy and reliability beyond doubt. The interviews lasted between 30–40 minutes and were recorded. To ensure accuracy, SY conducted triangulation with the assistance of a research assistant (RA). SY has extensive experience conducting qualitative research and has completed courses in qualitative research and good health research practice. She was also involved in a 2015 study on interprofessional collaboration. RA is a fresh graduate medical doctor who helps the investigator technically with data collection. All documents, recordings, and transcripts were made anonymous before being discussed within the research group. Colaizzi's seven-step phenomenological approach [46] was applied to the data analysis. The audio records of the interviews were transcribed manually in Indonesia by the research assistant within 3x24 hours after the interviews. The entire data analysis process was conducted in

**Table 2. Statistic literature of provider adherence to antenatal care guidelines.**

| Author, Source | setting and sample size | Study design | Summary of findings |
|---|---|---|---|
| Majrooh MA, Hasnain S, Akram J, Siddiqui A, Memon ZA. Coverage and quality of antenatal care provided at primary health care facilities in the "Punjab" province of "Pakistan." *PLoS One*. 2014;9(11). doi:10.1371/journal.pone.0113390 | 19 primary healthcare facilities of the public sector (seventeen Basic Health Units and two Rural Health Centers were randomly selected from each district | Mixed Method Quantitative and Qualitative methods | The overall enrollment for antenatal check-ups was 55.9%, and dropout was 32.9% in subsequent visits. The quality of services regarding assessment, treatment and counselling was extremely poor. The reasons for low coverage and quality were the distant location of facilities, deficiency of facility resources, indifferent attitude and non-availability of the staff. Moreover, a lack of client awareness about the importance of antenatal care and self-empowerment for decision-making to seek care was also responsible for low coverage. |
| Seyoum T, Alemayehu M, Christensson K, Lindgren H. Client Factors Affect Provider Adherence to Guidelines during First Antenatal Care in Public Health Facilities, Ethiopia: A Multi-Center Cross-Sectional Study. *Ethiop J Health Sci*. 2020;30(6):903–912. doi:10.4314/ejhs.v30i6.8 | Public Health Facilities, Ethiopia 834 study participants **Providers' adherence**: Providers' adherence was defined as providers' "conformity to, fulfilling standard ANC guidelines as per the national protocols | Cross-sectional study | The proportion of clients who received the complete provider's adherence to the first antenatal care guideline was 32.25% (95% CI: 29.1–35.5). The mean adherence score was 16.78%. Women who had a prior history of pregnancy and or birth-related complications (AOR = 1.58; 95%CI: 1.04–2.04) and late antenatal care booking at gestational week 16 or greater (AOR = 1.45; 95%CI: 1.03–2.03) were significantly associated with clients receiving complete providers' adherence to the first antenatal guideline |
| Benova L, Tunçalp Ö, Moran AC, Campbell OMR. Not just a number: Examining coverage and content of antenatal care in low-income and middle-income countries. *BMJ Glob Health*. 2018;3(2):1–11. doi:10.1136/bmjgh-2018-000779 | 10 LMICs analyse ANC related to women's most recent live birth up to 3 years preceding the survey | Descriptive Demographic and Health Survey | In all ten countries, the majority of women in need of ANC reported 1+ ANC visits and over two-fifths reported 4+ visits. Receipt of the six routine components varied widely; blood pressure measurement was the most commonly reported component, and urine test and information on complications the least. Among the subset of women starting ANC in the first trimester and receiving 4+ visits, the percentage of receiving all six routine ANC components measurement was low, ranging from 10% (Jordan) to around 50% in Nigeria, Nepal, Colombia and Haiti. Even among women with patterns of care that complied with global recommendations, the content of care was poor. |
| Mansur AMSA, Rezaul KM, Mahmudul HM, S C. Quality of antenatal care in primary health care centers of Bangladesh. *J Family Reprod Health*. 2014;8(4):175–181. | purposively selected three upazilas among the clients receiving antenatal care (ANC). Data were collected with a questionnaire cum checklist in the context of two aspects of quality issues, namely, assessment of physical arrangements for ANC (input) and services rendered by the providers (process). | descriptive cross-sectional | Out of 528 multigravid respondents, 360 (68.2%) took ANC during their previous pregnancy, whereas 168 (31.8%) did not take ANC. Pregnancy outcome was found to be associated with receiving ANC ($\chi2$ = 73.599; p = 0.000). Respondents receiving ANC had more good pregnancy outcomes. The mean waiting time for receiving ANC was 0.77±.49 hours. Out of the 13 centers, only 3 (23.1%) have sufficient instruments to render ANC services. Findings showed that the modes of ANC service delivery in the ANC centers are fairly satisfactory. Though some of the points of standard operation procedures (SOPs) on ANC are not covered by some ANC centers, those were not considered necessary. However, regarding the physical facilities available for rendering ANC services, it is seen that the facilities are not quite satisfactory. The number of doctors and nurses was not very satisfactory. One of the centers under this study has no doctor, where nurses give ANC services. More than 60% of the respondents received the services and advice. |

Indonesian. The two researchers (SY, HW) cross-checked the audio records and applied the seven-step approach for the data analysis: 1. To obtain a more accurate understanding of the descriptions, they read each transcript carefully and combined it with the field notes taken from the interviews. 2. They extracted meaningful statements related to the participant's perspective and the barriers of and promoting factors for the provider's adherence to national antenatal care guidelines. 3. They coded significant statements, labelled them with the participants' keywords and phrases, and cross-checked and discussed all the codes. 4. They repeated and checked the first three steps, and each code was read, evaluated repeatedly, and deliberately before clustering them into multidimensional categories. 5. They identified and described three sub-themes 6. They determined the final theme based on continuous discussion, comparison, reintegration, and inspection of the sub-themes. 7. They returned the transcript to participants to check their accuracy. HW is a Master of Public Health who has experience as a facilitator of qualitative study courses.

They coded significant statements manually, labelled them with the participants' keywords and phrases, and cross-checked and discussed all the codes. Analysis was adaptive, integrating thematic areas that researchers had generated with input process and output models as components of the health system. The consensus was reached through regular analysis discussions, and we had different opinions on including the code limited health provider as an imbalance resource or role responsibility, and we consulted this difference to the research supervisor (AU). Finally, we note that a limited number of health care providers are included in the theme of Imbalance resources. The flowchart for coding acquisition is shown in Fig 1:

## Result

There were 279 medical records of pregnant women receiving integrated Antenatal Care (ANC) between November 2018 and August 2019. Of these, 131(46.95%) women did not complete the ANC visits (four times). Therefore, only 149 women were included in the analysis. Most women had completed high school, with the youngest age of 19 years, the oldest of 39 years, and the average age of respondents was 27.59. In addition, most respondents

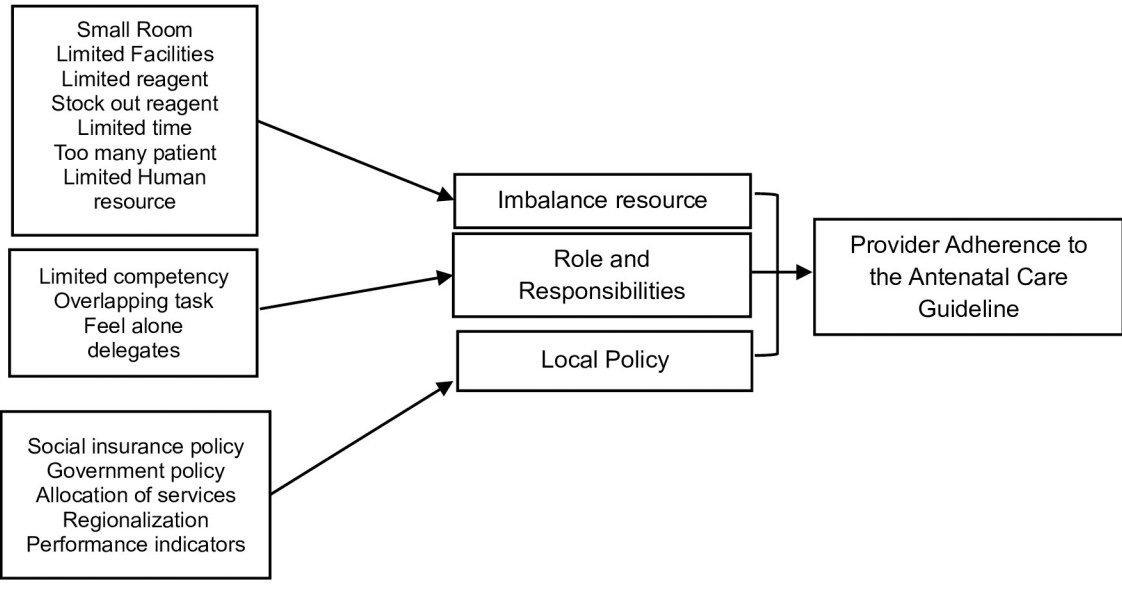

**Fig 1. Flowchart of thematic analysis.**

**Table 3. Demographic characteristics of pregnant women in four primary health center in Semarang, Indonesia.**

| Characteristics | Adherent to Integrated ANC guideline (n 47) | | Non-adherent to Integrated ANC guideline (n 102) | | p |
|---|---|---|---|---|---|
| | n | % | n | % | |
| Level of Education | | | | | |
| Junior high school | 10 | 21.3 | 12 | 11.8 | 0.104** |
| Senior high school | 35 | 74.5 | 82 | 80.4 | |
| Diploma | 0 | 0 | 1 | 1 | |
| Bachelor | 2 | 4.3 | 7 | 6.9 | |
| Age | | | | | |
| <20 years | 0 | 0 | 1 | 1 | 0.267* |
| 20–35 years | 44 | 93.6 | 87 | 85.3 | |
| >35 years | 3 | 6.4 | 14 | 13.7 | |
| Obstetric status | | | | | |
| Gravida | | | | | |
| 1 | 17 | 36.2 | 36 | 35.3 | 0.338** |
| 2 | 14 | 29.8 | 46 | 45.1 | |
| 3 | 13 | 27.7 | 18 | 17.6 | |
| 4 | 3 | 6.4 | 2 | 2 | |
| Parity | | | | | |
| 0 | 17 | 36.2 | 37 | 36.3 | 0.323** |
| 1 | 17 | 36.2 | 52 | 51 | |
| 2 | 13 | 27.7 | 12 | 11.8 | |
| 3 | 0 | 0 | 1 | 1 | |
| History of abortion | | | | | |
| 0 | 40 | 85.1 | 96 | 94.1 | 0.072** |
| 1 | 6 | 12.8 | 5 | 4.9 | |
| 2 | 1 | 2.1 | 1 | 1 | |
| Primary health center | | | | | |
| Basic Emergency Obstetric and Newborn Care | 30 | 63.8 | 43 | 42.2 | 0.332* |
| Non-Basic Emergency Obstetric and Newborn Care | 17 | 36.2 | 59 | 57.8 | |

BEmONC: Basic Emergency Obstetric and Newborn Care

*Contingency coefficient

**Spearman correlation test

experienced their first pregnancy (primigravida) and had no history of miscarriages. This study reported that age, level of education, obstetric status, and Primary health center (PHC) status In Semarang, Indonesia, were not associated with provider adherence to the integrated ANC guideline (Table 3).

Table 3 shows that there was a lack of adherence to antenatal care services in one pregnant woman under the age of 20 and 1 pregnant woman (1%) with more than three previous pregnancies. It is important to further investigate whether there are barriers such as shame or fear preventing pregnant women from undergoing antenatal care examinations as per national guidelines. Unfortunately, the researchers were unable to interview the patient, which hindered the exploration of the patient's experience with the service. Future research should include a more balanced representation of the age group under 20 and women with more than three previous pregnancies in order to gain a comprehensive understanding of antenatal care services within this demographic.

**Table 4. The demographics of in-depth interview participants.**

| Participant | Participant 1 | Participant 2 | Participant 3 | Participant 4 | Participant 5 |
|---|---|---|---|---|---|
| Age | 43 year | 48 year | 37 year | 38 year | 35 year |
| Working duration | 22 year | 24 year | 15 year | 17 year | 15 year |
| Workplace | PHC A | PHC B | PHC C | PHC D | PHD A |
| Title | midwife coordinator | midwife coordinator | midwife coordinator | midwife coordinator | Laboratory analyses |
| Employment status | permanent | permanent | permanent | permanent | permanent |

In-depth interviews were conducted with five healthcare providers who were involved in delivering integrated antenatal care. The aim was to explore the factors that promote or hinder provider adherence to the guidelines for integrated antenatal care. Participant demographics are presented in Table 4.

## Provider adherence

According to this study, only 6% of pregnant women received all recommended procedures for integrated antenatal care (ANC) per the Indonesian Ministry of Health and local government policy. The study also found no significant difference in healthcare providers' adherence to modified integrated ANC between two types of PHCs (BEmONC and non-BEmONC; p = 0.268). Both types of community PHCs failed to perform all integrated ANC procedures for all pregnant women visiting ANC. However, upon quantitative analysis, there was a difference in healthcare providers' adherence scores (p = 0.000) to the modified integrated ANC guidelines between BEmONC and non-BEmONC PHCs (Table 6). In Basic Emergency Obstetric and Newborn Care (BEmONC), a greater percentage of pregnant women (63.8%) received integrated ANC procedures compared to non-BEmONC (36.2%) (Tables 5 and 6)

According to the study, healthcare providers in BEmONC PHC adhered more to the modified national integrated antenatal care in most procedures (Table 5). However, the percentage of provider adherence to the mid-upper arm circumference examination procedure in BEmONC PHC (4.1%) was slightly lower than in non-BEmONC (7.9%). The general physical examination was the most commonly unrecorded procedure across the four ANC visits. More women in BEmONC (60.82%) received urine protein examinations twice or more during the pregnancy than non-BEmONC. In contrast, the provider adherence in haemoglobin examination was better in non-BEmONC (64 pregnant women (84.2%) received twice or more haemoglobin examinations during the period of pregnancy) than BEmONC (43 pregnant women (58.9%) received twice or more haemoglobin examinations during the period of pregnancy).

The BEmONC has better documentation on physical examination and laboratory examination findings, along with the four ANC visits (Table 6). The average score for the integrated ANC procedure was higher (42.26) compared to non-BEmONC (39.8).

## Barriers and enablers of provider adherence

The study has identified various factors that influence the adherence of healthcare providers to integrated antenatal care procedures. These include human resources, infrastructure, and policies implemented in health facilities, such as imbalance of resources, role and responsibilities issues, and local policy issues regarding funding for examinations in PHC. Furthermore, the study has found that resource differences can be observed in infrastructure and the number of workers. For instance, BEmONC PHC has more infrastructure and workers compared to non-BEmONC PHC. The number of general practitioners, midwives, patients and residents living in their working areas is two-thirds higher than non-BEmONC PHC (Table 7).

**Table 5. Number of examinations performed across four visits of integrated Antenatal Care in four Primary health center in Semarang, Indonesia.**

| Integrated antenatal care minimum package examination (number of performed examinations) | BEmONC (n = 73) n(%) | Non-BEmONC (n = 76) n(%) | TOTAL (n = 149) n(%) |
|---|---|---|---|
| General physical examination | | | |
| 0 | 9 (12.3) | 18 (23.7) | 27 (18.1) |
| 1 | 4 (5.5) | 0 (0.0) | 4 (2.7) |
| 2 | 0 (0.0) | 29 (38.2) | 29 (19.5) |
| 4 | 60 (82.2) | 29 (38.2) | 89 (59.7) |
| Temperature, Blood pressure, Body mass | | | |
| 4 | 73 (100) | 76 (100) | 149 (100) |
| Mid-upper arm circumference | | | |
| 1 | 70 (95.9) | 70 (92.1) | 140 (94) |
| 4 | 3 (4.1) | 6 (7.9) | 9 (6.0) |
| Height of fundus uteri, Fetal presentation, and Fetal heart rate | | | |
| 3 | 73 (100) | 76 (100) | 149 (100) |
| Hemoglobin test | | | |
| 1 | 30 (41.1) | 12 (15.8) | 42 (28.2) |
| 2 | 13 (17.8) | 44 (57.9) | 57 (38.3) |
| 3 | 0 (0.0) | 3 (3.9) | 3 (2.0) |
| 4 | 30 (41.1) | 17 (22.4) | 47 (31.5) |
| Blood type test | | | |
| 1 | 13 (17.8) | 46 (60.5) | 59 (39.6) |
| 2 | 57 (78.1) | 24 (31.6) | 81 (54.4) |
| 4 | 3 (4.1) | 6 (7.9) | 9 (6) |
| Urine protein level test | | | |
| 1 | 13 (17.8) | 45 (59.2) | 58 (38.9) |
| 2 | 0 (0.0) | 2 (2.6) | 2 (1.3) |
| 3 | 30 (41.0) | 12 (15.8) | 42 (28.2) |
| 4 | 30 (41.0) | 17 (22.4) | 47 (31.5) |
| Blood glucose/reduction test | | | |
| 3 | 13 (17.8) | 47 (61.8) | 60 (40.3) |
| 4 | 60 (82.2) | 29 (38.2) | 89 (59.7) |
| Hepatitis and HIV serology test | | | |
| 1 | 73 (100) | 76 (100) | 149 (100) |
| Malaria, Acid-Fast Bacilli (AFB), Syphilis test, and Ultrasonographic examination | | | |
| 0 | 73 (100) | 76 (100) | 149 (100) |
| Interdisciplinary examination and education | | | |
| Midwives, General Practitioner, and Nutritionist | | | |
| 4 | 73 (100) | 76 (100) | 149 (100) |
| Dentist | | | |
| 0 | 68 (93.2) | 58 (76.3) | 126 (84.6) |
| 1 | 2 (2.7) | 12 (15.8) | 14 (9.4) |
| 4 | 3 (4.1) | 6 (7.9) | 9 (6) |

BEmONC: Basic Emergency Obstetric and Newborn Care, PHC: Primary Health

## Imbalance of resources

Several participants stated that the main obstacles to integrated ANC were limited time, a high number of patients, a shortage of staff such as midwives and physicians, and insufficient

**Table 6. Fulfilment of ANC examination criteria per visit in four primary health center in Semarang, Indonesia.**

| | BEmONC PHC(n 73) | | Non-BEmONC PHC(n 76) | | p |
|---|---|---|---|---|---|
| | Adherent n (%) | Non-adherent n (%) | Adherent n (%) | Non-adherent n (%) | |
| ANC Examination Score | | | | | |
| Visit 1 | | | | | |
| Physical Examination | 64 (87.7) | 9 (12.3) | 29 (38.2) | 47 (61.8) | <0.00* |
| Laboratory Examination | 73 (100) | 0 (0.0) | 76 (100) | 0 (0.0) | - |
| Multidisciplinary examination | 3 (4.1) | 70 (95.9) | 6 (7.9) | 70 (92.1) | 0.268** |
| Visit 2 | | | | | |
| Physical Examination | 60 (82.2) | 13 (17.8) | 53 (69.7) | 23 (30.3) | 0.076* |
| Laboratory Examination | 60 (82.2) | 13 (17.8) | 29 (38.2) | 47 (61.8) | <0.00* |
| Multidisciplinary examination | 73 (100) | 0 (0.0) | 76 (100) | 0 (0.0) | - |
| Visit 3 | | | | | |
| Physical Examination | 60 (82.2) | 13 (17.8) | 34 (44.7) | 42 (55.3) | <0.00* |
| Laboratory Examination | 30 (41.1) | 43 (58.9) | 17 (22.4) | 59 (77.6) | 0.014* |
| Multidisciplinary examination | 73 (100) | 0 (0.0) | 76 (100) | 0 (0.0) | - |
| Visit 4 | | | | | |
| Physical Examination | 60 (82.2) | 13 (17.8) | 58 (76.3) | 18 (23.7) | 0.377* |
| Multidisciplinary examination | 73 (100) | 0 (0.0) | 76 (100) | 0 (0.0) | - |
| ANC guideline adherence | | | | | |
| 1st Visit | 3 (4.1) | 70 (95.9) | 6 (7.9) | 70 (92.1) | 0.268** |
| 2nd Visit | 60 (82.2) | 13 (17.8) | 29 (38.2) | 47 (61.8) | <0.00* |
| 3rd Visit | 30 (41.1) | 43 (58.9) | 17 (22.4) | 59 (77.6) | 0.014* |
| 4th Visit | 60 (82.2) | 13 (17.8) | 58 (76.3) | 18 (23.7) | 0.377* |
| Overall ANC visit | 3 (4.1) | 70 (95.9) | 6 (7.9) | 70 (92.1) | 0.268** |
| Overall ANC visit without dentist visit 1 | 30 (41.1) | 43 (58.9) | 17 (22.4) | 59 (77.6) | 0.014* |
| ANC guideline adherence score | | | | | |
| Maximum score | 45 | - | 45 | - | <0.000*** |
| Minimum score | 36 | - | 36 | - | |
| Average | 42.26 | - | 39.8 | - | |

BEmONC: Basic Emergency Obstetric and Newborn Care, PHC: Primary Health

*Chi-square

** Fisher's exact test

***Mann-Whitney U test

diagnostic reagents, particularly for hepatitis B and HIV tests. To obtain a comprehensive overview, we cross-referenced the opinions gathered with information on human resources and the daily patient count at each PHC.

In most cases, BEmONC PHC facilities have more doctors and midwives than non-BEmONC facilities. This is because BEmONC PHCs offer a wider range of services and tasks. However, even though the number of health professionals in each PHC meets the Ministry of Health standards, there is a shortage of providers who can deliver integrated ANC. This may be due to logistical issues such as a lack of laboratory examination reagents and manual recording of reagent usage, which can lead to reagent stock-outs. The fact was stated in the following quote:

**Table 7. Characteristics of the primary health centers (PHC) research site.**

| Characteristics | PHC A | PHC B | PHC C | PHC D |
|---|---|---|---|---|
| Number of Physicians | 4 | 4 | 2 | 5 |
| Number of midwives | 11 | 11 | 4 | 5 |
| Number of Lab Analysts | 2 | 1 | 1 | 3 |
| Number of nutritionists | 1 | 1 | 1 | 1 |
| Number of dentists | 1 | 1 | 1 | 2 |
| Total number of patients per day | 100–130 | 100–130 | 100–120 | 70–100 |
| Number of Antenatal Care Visit per day | 35 | 35 | 20 | 20 |
| Basic Emergency Obstetric and Newborn Care | Yes | Yes | No | No |
| Population | 33,239 | 67,541 | 40,336 | 41,749 |
| Number of Women in reproductive age | 9,533 | 20,000 | 12,286 | 13,647 |

Source: secondary data from the Primary health center and Semarang population database.

"We roughly estimate the number of reagents needed, and we will order the new reagent if the reagent remains 25%. I counted all the reagent expenditures every day. Sometimes, if there were many patients, I would delay it. Tomorrow morning, I will write it down in this book."

(Female analyst on non-BEmONC PHC 4)

The laboratory analysts complained about the problem with the inventory of reagents, especially on inventory control. They complain that manual inventory of reagents leads to inaccurate supply data, affecting the availability of the stock and creating a shortage of diagnostic reagents for antenatal care examination. Field observation showed that the laboratory care demand was 20 to 35 samples daily, run by one analyst with 7–8 working hours daily.

## Role and responsibilities issues

The midwives, the foremost provider of integrated ANC in the two types of PHC, complained that they provided the ANC on their own, without any assistance from others. They handled patients with anamnesis, obstetric examination, and education. They also take other health professionals' duties in exceptional conditions, including physical examination and medical treatment.

"The patient will be managed first by the midwife, and she will examine the vital the body weight of the patient, and others examination. Later, if we find a new risk, we will then consult the doctor"

(Midwife PHC 1)

"We will handle the education if other providers can not do that for any reason "

(midwife PHC 2)

The lack of physician involvement was associated with the little MCH room and their overworkload. Two midwives mention that they sometimes ask the doctor to put a signature on the medical record to comply with the integrated ANC guideline, even though the doctor does not perform an integrated ANC examination.

"Yes, frequently there was only one doctor provided medical services, sometimes if we see a lot of patients in the general clinic, we only offered the doctors to sign the medical records even they did not examine and give education to the patient"

(Midwife PHC 3)

One midwife describes that the general physical examination was not their role, as mentioned in the following statement:

"Yeah, if we do a general physical examination, we don't know that deeply; that's not our knowledge, we just know a little, hehe

(PHC 4 midwife)."

However, in BEmONC PHC, midwives and physicians collaborated to provide the integrated ANC in the Maternal and Child Health (MCH) room, while in non-BEmONC, they performed this task separately. The MCH room in BEmONC is larger than that in non-BEmONC. Therefore, it facilitates collaborative care between doctors and midwives in delivering integrated ANC. In BEmONC, the Physician involvement was motivated by the midwife coordinator as described in the following way:

"I told them (physician). Anyway, I would not have started an integrated ANC examination if the doctor was not present at the MCH clinic"

(midwife coordinator of PHC 1)

Midwives in BEmONC PHC said that the doctors and midwives had been trained collaboratively on maternal and neonatal emergency management and received additional training from CEmONC hospital every three or six months. Therefore, they are more familiar with collaborative services.

## Local policy issue

The respondent complained about a high workload, and this was related to the local government policy, as follows:

"Sometimes pregnant women outside our responsibility area come, we deliver the integrated ANC, but we can not report it in our performance indicators. It is just additional work. We send the result of the integrated ANC to the appropriate PHC. Can you imagine if so many women do that?

(midwife coordinator in PHC B) "

She also mentioned that the high workload was due to the patient's lack of knowledge of the regionalisation policy. In addition, it was partly due to the local government's lack of adherence to health service regionalisation enacted by the Health Social Security Administrative Bodies (Badan Penyelenggara Jaminan Sosial Kesehatan /BPJS-K). The local government policy stated that every citizen of Semarang could access health services throughout PHC in Semarang for free, regardless of the regionalisation rules of primary health facilities set by the BPJS-K.

We triangulated the midwife's opinion with the policy document review. There was a contradicting policy on the regionalisation of PHC among local governments and the BPJS-K. The

local governance provides a free integrated ANC in all PHCs for pregnant women with Semarang identity cards without regionalisation. At the same time, BPJS-K limited the service performance assessment of PHC, especially the contact rate indicator (the number of members of Indonesian Health Social Insurance visited a health facility or visited by the health provider), only counted the services the patients who already registered as a member of BPJS-K in the PHC.

## Discussion

This study found complete adherence to the integrated ANC guideline level of 6% amongst health providers in the two types of PHC. These findings were relatively lower than the findings of most studies conducted in Jakarta (14.6%) [19], West Java (68%) [30], Ghana (48%) [26], Malaysia (48%) [47], and but slightly higher than study in Nigeria (4.6%) [48]. Various findings were linked to the differences in assessment techniques, including directly observing patients, reviewing medical records, and analysing the content of integrated ANC procedures across different regions. Statistically, the total adherence level to the integrated ANC guideline was not different between health providers in BEmONC and non-BEmONC. The results of this study are affected by the number of samples and the limitations of the chi-square statistical test that was used. This test is very sensitive to sample size and is more suitable for studies with a large number of samples. In this particular study, multiple sample categories have been combined to ensure a larger sample size, but this may not completely overcome the limitations of the chi-square test. As a result, a larger sample size is needed in the next study in order to effectively compare the services offered by the two types of Community Health Centers. This will ensure that the expected frequency in each group is more than 5, allowing for the use of the chi-square test [49–51].

However, Provider adherence was higher in the non-BEmONC in most ANC procedures, namely anamnesis, physical examination, laboratory examination, and interdisciplinary assessment. The blood type examination was identified as an inefficient laboratory examination. The study revealed that healthcare providers in Semarang are not strictly adhering to the national integrated antenatal service guidelines. This is quite unexpected, especially since antenatal care coverage is as high as 98%. It is, therefore, crucial to reevaluate the criteria used to measure the success of ANC implementation, not just focusing on the quantity of care provided but also on its quality. These findings further support the research conducted in Tanzania, which revealed that the provision of ANC services was inconsistent and did not align with the recommended Focus Antenatal Care (FANC) guidelines [52].

This study also reported poor documentation of integrated ANC, mainly the dentist visits and the physical examination. The medical record review showed excellent documentation of interdisciplinary examination and education, except for the first visits. In addition, the ANC form does not mention general physical examination, which causes the health workers to forget to write down the results of medical check-ups in the medical record. Still, the interdisciplinary education was excellent in all PHCs except for the dentist visit on the first visit. These findings indicate that dentists should be more involved in integrated antenatal care. This is supported by a study conducted in Jeddah, which found that only 33.2% of women had their oral health checked during pregnancy [53]. The study in Jakarta reported that the height measurement result was not recorded in all ANC documents. Only 30% and 9–11% of the diameter of the upper arm and laboratory test were recorded, respectively [19]. A different finding from the existing literature was an excellent record of interdisciplinary management in all PHCs, proved by the provider assignment in the medical record. However, patient management was usually done only by midwives. The other professionals, especially physicians, only gave an

assignment in the medical record. A study conducted in 10 low to middle-income countries (LMIC) revealed that health providers, particularly physicians, had limited knowledge about pregnancy complications [8]. Existing literature showed that limited human resources in terms of number and competency are the barriers to delivering full coverage integrated ANC content. The limited number of physicians was the barrier to their involvement, resulting in non-adherence to the integrated ANC guideline, especially on general physical examination procedures. It is also essential to include a checklist of physical examination results in the patient's medical record. This checklist should contain the outcomes of all integrated antenatal care examination procedures to minimise unrecorded examinations by health workers.

Regulation Of The Republic Of Indonesia Health Minister number 97 in 2014 emphasises the provisions and authority of integrated ANC examination in both BEmONC and non-BEmONC PHC [29]. Although the PHC resource met the standard enacted by the Ministry of health [54], the number of health providers, MCH room capacity, and data management seem inadequate to provide an ideal integrated ANC, especially in non-BEmONC PHC. In general, the BEmONC PHC has more doctors and midwives than non-BEmONC. It is related to the services provided by BEmONC PHC, including basic maternal and neonatal emergency management. However, concerning the integrated ANC services, BEmONC and Non-BEmONC PHC have the same responsibility of providing a high-quality integrated ANC [29, 30]. Both PHCs had five or more midwives who were not solely responsible for delivering ANC services annually. Several low- and middle-income countries (LMICs) have faced serious incidents hindering the implementation of antenatal care (ANC). For instance, in Tanzania, some health facilities have issues such as mouldy walls, damaged roofs and windows, and bird nests inside the facility. Additionally, many facilities lack access to water and electricity, frequently run out of essential medicines, and do not have maternal and child health cards or books to record ANC visit history and examination results [55].

This study found the role and responsibilities issue between doctors and midwives promotes unrecorded general physical examination results. Midwives were not accustomed to recording the results of general physical examinations, although they have done it. The general physical examination record was more relevant to be carried out by general practitioners. The physician's involvement in integrated ANC could improve health services quality, including effectiveness, efficiency, and accuracy of pathological conditions in pregnant women. A study in Sweden revealed that unclear allocation of responsibilities is a barrier to collaborative practice among health professionals from different organization [56]. The unclear role and responsibilities lead to an imbalance of shared workload and overwhelmed feelings among the foremost health care providers [57]. Since the midwives were the major providers in integrated ANC, they needed collaborative work with other health professionals to reduce missed opportunities, perform early diagnosis and prompt treatment. Better collaboration care among doctors and midwives in BEmONC is promoted by regular collaborative training and experience. Studies in Rwanda also indicate the need to improve room facilities, availability of competent health providers, and interprofessional collaboration to enhance the quality of integrated ANC implementation [58].

The barriers to provider adherence in the laboratory test were the reagent limitation, lack of strategic management and supervision, and continuous improvement [59]. The study found that poor laboratory procurement was related to manually documented reagent utilisation. It promotes a false count of stock and causes a reagent stock out. Based on the existing studies in Indonesia and low-income countries, the limitation of laboratory services was a barrier to the provider's adherence to integrated ANC [60]. According to a survey conducted in Jakarta, laboratory staff faced an overwhelming workload, and the absence of analysts (due to illness or leave) resulted in incomplete examinations of pregnant women during their second trimester

[19, 61]. Pharmacy officers do not fully understand how to use various applications as tools to carry out stocktaking more effectively and efficiently, so they still rely on manual methods. Despite the mandatory paperless policy and electronic medical records introduced by the head of the Semarang Health Service, it is essential to conduct evaluations. This includes assessing the readiness of the officers, providing regular training, and developing the system based on feedback and suggestions from those implementing it.

The regionalisation of PHC in delivering the integrated ANC should be considered in a local government policy to prevent maldistribution of the patient and give the provider enough time to deliver all procedures of an integrated ANC. For illustration, PHC had 120–150 patients per day within 5 to 6 work hours, so it is estimated that the patient will meet the doctor in 7 to 10 minutes. Meanwhile, complete adherence to the integrated ANC guideline takes at least 4 hours for each patient, including the laboratory test, interdisciplinary examination and education. An additional patient will prolong waiting time and hinder the health provider's adherence to the integrated ANC guideline. The issue of regionalisation and uneven distribution of healthcare workers is also evident in several low- and middle-income countries (LMICs). For example, in rural districts of Tanzania, there are more than 30–50 new first antenatal care (ANC) visits per month. Many of these women come from outside the local area, travelling from neighbouring districts and surrounding wards that have limited health facilities [55].

The existing literature reveals that despite the regionalisation of the health facility program's improved collaboration and communication, there is a potential conflict among federal, state, and local government responsibilities that will inhibit the public health response to other public health emergencies. However, other study reveals that it improved public health preparedness and surveillance [62]. This study indicates that regionalisation should be applied to localise and maximise resource utilisation. It is important to establish clear guidelines for cooperation between the city leadership of Semarang and the leadership of BPJS-K to implement the policy of regionalising health services. These guidelines should ensure that the community can access services according to their needs without being restricted by regional boundaries. It's crucial to avoid overburdening specific health facilities while meeting the public's demand for services. Additionally, equitable development of health facilities throughout Semarang is necessary to ensure that high-quality healthcare is accessible to the entire community.

This study suggested the importance of inter-health centers collaborative care in delivering the integrated ANC, which was promoted by each health professional's clear role and responsibility. In addition, local government policy should follow the national integrated ANC guidelines to promote the implementation of integrated ANC. Like most qualitative data, the findings are generalisable only to the primary health center of interest in the region of urban Indonesia. Unfortunately, this study did not explore integrated ANC guidelines' perception and adoption factors among physicians and other professions. However, the information is essential to improve the integrated ANC. Hence, a study concerning those factors is needed.

In terms of limitation, the research findings may not be generalisable due to the sensitivity of the statistical tests to sample size [49]. This study included a sample of four PHCs collaborating with the hospital where the principal researcher works, but it is, however, important to note that the selected PHC represent the characterise of the two different types of PHC (BEmONC and non-BEmONC) and two different areas (Suburban and urban). Due to the fact that we only evaluated the adherence based on the medical record, there may be a problem with the accuracy of the data; for example, the provider may not always record the procedure that they performed in the medical record. Further studies may need to evaluate the adherence based on observation.

## Conclusion

Although the integrated ANC guideline was enacted by the Indonesian Ministry of Health in 2014, few women in the four PHCs In Semarang, Indonesia (6%) received the full content of the integrated ANC. The average score of provider adherence to the integrated ANC guideline in non-BEmONC PHC (42,26) was higher than BEmONC (39,8). However, there was no statistical difference (p = 0.268). The barriers to provider adherence were an imbalance of resources, roles, responsibilities, and local policy issues. It is essential to provide collaborative training for midwives, dentists, and general practitioners on integrated antenatal care procedures and pregnancy complication screening. This will ensure that multi-professional services can be effectively carried out in both types of health centers. When scheduling the training, it's important to consider the service hours at the health center in order to maximise the participation of invited health workers, especially at non-Basic Emergency Obstetric and Newborn Care (BEmONC) health centers, without disrupting service hours. It is also important to conduct longitudinal research to assess officer compliance after implementing various recommendations from this study. This includes ensuring infrastructure fulfilment, improving the format of ANC patient medical records, providing training on procedures for implementing collaborative and integrated antenatal care and aligning health service regionalisation policies. This will enable continuous improvement in integrated antenatal care services to meet the high-quality health needs of pregnant women later on. This study recommends the need for regionalisation of PHC work area, authority within primary care staff, and also intra- and inter-health centers collaborative care to provide high-quality integrated antenatal care for every woman in Indonesia.

## Supporting information

**S1 Checklist. Ochatain mixed method checklist for integrated ANC study.**
(DOCX)

**S1 File. Integrated ANC analysis SPSS.**
(XLSX)

**S2 File. Integrated ANC data.**
(XLSX)

## Acknowledgments

The authors acknowledge the respondents and all the four primary health center in Semarang who participated in this study, Universitas Gajah Mada and Universitas Islam Sultan Agung, for facilitating this study.

## Author Contributions

**Conceptualization:** Suryani Yuliyanti, Adi Utarini, Laksono Trisnantoro.

**Data curation:** Suryani Yuliyanti.

**Formal analysis:** Suryani Yuliyanti.

**Funding acquisition:** Suryani Yuliyanti.

**Methodology:** Suryani Yuliyanti, Adi Utarini.

**Project administration:** Suryani Yuliyanti.

**Resources:** Suryani Yuliyanti, Adi Utarini.

**Supervision:** Suryani Yuliyanti, Laksono Trisnantoro.

**Validation:** Suryani Yuliyanti, Adi Utarini.

**Writing – original draft:** Suryani Yuliyanti, Adi Utarini, Laksono Trisnantoro.

**Writing – review & editing:** Suryani Yuliyanti, Adi Utarini.

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
