## [Decision Letter · Decision Letter 0]

26 Jun 2023

PONE-D-23-12582A mixed-method analysis of provider adherence to integrated antenatal care guideline in primary health centre: An Indonesian casePLOS ONE

Dear Dr. Yuliyanti,

Thank you for submitting your manuscript to PLOS ONE. After careful consideration, we feel that it has merit but does not fully meet PLOS ONE’s publication criteria as it currently stands. Therefore, we invite you to submit a revised version of the manuscript that addresses the points raised during the review process.

We look forward to receiving your revised manuscript.

Kind regards,

Phyllis Lau, PhD

Academic Editor

PLOS ONE

Reviewers' comments:

Reviewer's Responses to Questions

**Comments to the Author**

1. Is the manuscript technically sound, and do the data support the conclusions?

Reviewer #1: Partly

Reviewer #2: Yes

2. Has the statistical analysis been performed appropriately and rigorously? 

Reviewer #1: No

Reviewer #2: Yes

3. Have the authors made all data underlying the findings in their manuscript fully available?

Reviewer #1: No

Reviewer #2: Yes

4. Is the manuscript presented in an intelligible fashion and written in standard English?

Reviewer #1: No

Reviewer #2: Yes

5. Review Comments to the Author

Reviewer #1: Abstract.

Please describe where and when the study was conducted. How do you assess the adherence?

Introduction.

Please provide the full name of CEmNOC before put the acronym.

This sentence should be place in the methods: "Health providers in BEmNOC PHC administer parenteral drugs (i.e., antibiotics, uterotonic, oxytocin, and magnesium sulfate for preeclampsia and eclampsia). They also manually remove the placenta and retained products (e.g. manual vacuum extraction, dilation, and curettage), assisted vaginal delivery, and basic neonatal resuscitation."

Methods.

The 1st and 2nd paragraph of study setting should be part of the introduction as the rationale of the study.

Table 1 should be part of the result.

There some redundancy of information between the study design section and the quantitative and qualitative section. Please select the information that you put there carefully, without repetition.

Why do you choose medical records of ANC visits from November 2018 and August 2019 and not a 1-year period instead?

You stated that you select the PHC that collaborates with the hospital where the principal researcher works. Isn't that cause selection bias? How do you managed it?

You stated that "The provider adherence to integrated ANC guidelines was determined using the Indonesian Ministry of Health indicators enacted in 2012." However, in the result you also mentioned that you also analyzed whether it adhere to the local policy. Please also stated it in the methods.

Please give more detail about the type of socio-demographic data collected.

In the results, besides Kruskal Wallis, you also use Chi-square and Fisher. Please stated it in the methods, and explain which data use which type of analysis. Also you must stated here that you analyzed the average for the continuous data, and conduct statistical analysis to identify the difference. Have you conducted normality test for numerical data?

Please give more description regarding the SPSS, the company that produce it and the city where the company is located.

Please also give more detail information regarding the "observation". Who conducted it? What is the instrument utilized?

What is the qualification of the interviewer on the qualitative part? Since you stated that the PHCs were conveniently selected, how do you ensure that there is no interviewer bias?

Results.

The sentence, "These results showed that a compulsory education program and the minimum age of marriage law had been implemented well in this city22,33,34. It is belong to the discussion, not results.

Table 1. The term "study participants" is not appropriate, as you only analyzed the medical records which is not requiring participation of the women. Please also use complete full name for PHC and BEMNOC (not the acronym) and where is it located. Although it is already narrated previously. A table should be self explained.

The statement, "However, more pregnant women received an integrated ANC procedure in BeMNOC (average score of 42.26) than non-BEmNOC (average score 39.8)." This statement here reflect a proportion comparison, while you actually comparing the score, please rephrase.

The statement, "The medical record review showed excellent documentation of interdisciplinary examination and

education except for the first visits." This is actually an interpretation, which should be part of discussion. The result should only present fact.

Table 3. You use "general state" while in the methods you use term "general condition". Please be consistent.

Please use complete word for "and" not symbol "&". It appears in several parts of the manuscript.

For the p-value, please write "<0.001" for "0.00". The value is actually not zero, but the SPSS shortened it.

The sentence, "Still, the interdisciplinary education was excellent in all PHC except for the dentist visit in the first visits." This is also discussion.

You mentioned "reagent" several times. Please explain "reagent" for what.

In the quote line 277, what is "general checking" refers to? Is it the same meaning with "general condition" or "general status"? Please be consistent in the terminology.

You mentioned about policy document review. Please give more description in the methods.

Is BPJS-K the same with BPJS? Please be consistent.

Discussion.

The statement (line 317), The different findings were related to the difference in assessment methods and content of the integrated ANC procedure. Explain more about the difference in the assessment methods.

Line 327: LMIC (Low Middle Income Country), should be the complete name first and the acronym in the bracket.

Line 333: Ministry of Health Regulation No. 97 of 2014, did you mean the Ministry of Health of Indonesia?

Line 343: general perception assessment. what is "general perception assessment"? Please use same terminology through-out. Or provide explanation for each in the methods section.

You should acknowledge the biases in your study methods as the limitations.

How is the generalizability of this study?

Conclusion.

For the first statement please make clear that it is in Indonesia.

The statement, "The average score of provider adherence to the integrated ANC guideline in non-BEmNOC PHC (66,66) was higher than BEmNOC (33,3). However, there was no statistical difference (p = 0.000)." It seems to not appear in the result previously. Conclusion should be based on the result. Where is the statement appear in the result? Please don't add new result in the discussion or conclusion.

Acknowledgement.

Is The Indonesia Endowment Fund for Education funding the study? If so, please mention in the funding section.

General comment.

The manuscript need to be proof-read by an English language editor. Also the author mentioned about reporting guidelines, but it is not attached.

Reviewer #2: Dear Editor,

Thank you very much for the opportunity to review this manuscript, that the topic is close to my research interest and geographic expertise.

The authors have provide quite comprehensive study about the adherence of antenatal care in Semarang Indonesia.

From me, there are couple of minor revisions that the authors need to address before the publication of this paper:

1. The author mentions BemNOC many times, which is I understand, However, I would suggest to use a more widely used Basic Emergency Obstetric and Newborn Care (BEmONC)-See UNFPA publication, to represent Puskesmas with PONED-an Indonesian term for it.

2. Couple of abbreviations are not mentioned earlier but at the later part of the manuscript, for example LMIC-should be expand for the first time it is mentioned and BPJS-BPJSK.

3. Details for data analysis, data collection, trustworthiness, demographic of qual participants for the qualitative interviews/FGs are missing.

4. I would comment more also for the discussion and limitations, including implications for further research. This research would imply many, such as the need of training for non BEmONC, authority within primary care staff, and also perhaps further training for pregnancy complication screening-This was discussed superficially and authors discussed more on the numbers only.

5. Implication for further research and limitation of the study, including limited number of women and health providers involved in the study have not been clearly acknowledged. While Semarang is one of the big cities in Indonesia that usually the availability at least one doctor in the daily PHC practice is usually warranted. It raised concerns on the quality of care rather than numbers.

Thank you very much.

6. PLOS authors have the option to publish the peer review history of their article (what does this mean?). If published, this will include your full peer review and any attached files.

Reviewer #1: No

Reviewer #2: No

---

## [Author Response · Author response to Decision Letter 0]

19 Aug 2023

Dear Editor in chief,

Thank you for the opportunity to revise our manuscript, entitled: “A mixed-method analysis of provider adherence to integrated antenatal care guideline in primary health centre: An Indonesian case”. 

We do apologize for being late in manuscript revision and fulfill the editorial requirement. We appreciate the constructive suggestions given by the reviewers and we have made our best attempts to provide responses to the reviewers. 

In attach to this cover letter, please find our responses to the editor and two reviewer comments in the sentence colored blue. Changes made in the manuscript were marked using track changes as requested by editor. The revision has been developed in consultation with all coauthors, and each author has approved the final form of this revision. The agreement form signed by each author remains valid.

Sincerely,

Suryani Yuliyanti

---

## [Decision Letter · Decision Letter 1]

21 Nov 2023

PONE-D-23-12582R1A mixed-method analysis of provider adherence to integrated antenatal care guideline in primary health centre: An Indonesian casePLOS ONE

Dear Dr. Yuliyanti,

Thank you for submitting your manuscript to PLOS ONE. After careful consideration, we feel that it has merit but does not fully meet PLOS ONE’s publication criteria as it currently stands. Therefore, we invite you to submit a revised version of the manuscript that addresses the points raised during the review process.

**ACADEMIC EDITOR: **- If the aim of the study is to compare BeMONC and non-BeMONC centers, please indicate so in the title- The authors mentioned that this is a larger study with PHCs caring for women with heart diseases. Does this mean this study includes only women with heart diseases? Please elaborate**- "**Adherence to guidelines was assessed based on the fulfillment of items in the modified national integrated ANC guideline checklist"  What is the specific criteria? Does the failure of not checking one item is already considered non-adherence?- What is the estimated sample size for the qualitative phase? Why were only five women included?- What is the MMR in Semarang over time compared to the national level?- In Line 464, the conclusion mentions no statistical difference despite the p value being less than 0.000 (which cannot be true anyway). Please explain- In Line 242, the authors mentioned there were no differences, when it is more accurate to state that there is no statistically significant differences, as there are actually differences in the percentages.- I suggest the authors to use a proofreading service as there are careless typos all over the manuscript. The most egregious one is in line 123 with "agshsus", which I could not make neither heads nor tails what that mean.

We look forward to receiving your revised manuscript.

Kind regards,

Gilbert Sterling Octavius

Academic Editor

PLOS ONE

Reviewers' comments:

Reviewer's Responses to Questions

**Comments to the Author**

1. If the authors have adequately addressed your comments raised in a previous round of review and you feel that this manuscript is now acceptable for publication, you may indicate that here to bypass the “Comments to the Author” section, enter your conflict of interest statement in the “Confidential to Editor” section, and submit your "Accept" recommendation.

Reviewer #2: (No Response)

Reviewer #3: All comments have been addressed

2. Is the manuscript technically sound, and do the data support the conclusions?

Reviewer #2: Yes

Reviewer #3: Yes

3. Has the statistical analysis been performed appropriately and rigorously? 

Reviewer #2: Yes

Reviewer #3: Yes

4. Have the authors made all data underlying the findings in their manuscript fully available?

Reviewer #2: Yes

Reviewer #3: Yes

5. Is the manuscript presented in an intelligible fashion and written in standard English?

Reviewer #2: Yes

Reviewer #3: No

6. Review Comments to the Author

Reviewer #2: Unfortunately, it is still unclear to me in this version, as only one highlight in the manuscript so I don't really know which one has been edited. Could the author provide the tracked changes all of the edits? or highlight it?

At the response to the reviewers also do not specifically answering the reviewers comment.

Reviewer #3: The study reaches its aim to compare the provider’s adherence to the integrated antenatal care between two types of health care units. The structure is interesting but not innovative. Yet, maternal mortality reflects the quality of a health care system and therefore I have a positive opinion on this study.

Positive points:

Study design and methods are accurately described. The Statistical analysis used was adequate. Tables are clear and readable, table captions are complete and accurate.

Negative points:

The literature statistics should be in provided in table for easier studying.

I suggest a more detailed description of the local health care system. The structure of the health care system in developing countries and a brief description of the way it functions should be addressed.

In study design, lines 111-115 should be addressed in the results.

Language editing is recommended.

7. PLOS authors have the option to publish the peer review history of their article (what does this mean?). If published, this will include your full peer review and any attached files.

Reviewer #2: No

Reviewer #3: No

---

## [Author Response · Author response to Decision Letter 1]

29 Jan 2024

Dear Editor in chief,

Thank you for the opportunity to revise our manuscript, entitled: “A mixed-method analysis of provider adherence to integrated antenatal care guideline in BEmONC and Non BEmONC primary health center: An Indonesian case”. 

We do apologize for being late in manuscript revision and fulfill the reviewer requirement. We appreciate the constructive suggestions given by the reviewers and we have made our best attempts to provide responses to the reviewers. 

In attach to this cover letter, please find our responses to the editor and two reviewer comments in the sentence colored blue. Changes made in the manuscript were marked using track changes or highlight as requested by editor. The revision has been developed in consultation with all coauthors, and each author has approved the final form of this revision. The agreement form signed by each author remains valid.

---

## [Decision Letter · Decision Letter 2]

24 Jun 2024

PONE-D-23-12582R2A mixed-method analysis of provider adherence to integrated antenatal care guideline in BEmONC and Non BEmONC primary health center: An Indonesian casePLOS ONE

Dear Dr. Yuliyanti,

Thank you for submitting your manuscript to PLOS ONE. After careful consideration, we feel that it has merit but does not fully meet PLOS ONE’s publication criteria as it currently stands. Therefore, we invite you to submit a revised version of the manuscript that addresses the points raised during the review process.

We look forward to receiving your revised manuscript.

Kind regards,

Gilbert Sterling Octavius

Academic Editor

PLOS ONE

Journal Requirements:

Reviewers' comments:

Reviewer's Responses to Questions

**Comments to the Author**

1. If the authors have adequately addressed your comments raised in a previous round of review and you feel that this manuscript is now acceptable for publication, you may indicate that here to bypass the “Comments to the Author” section, enter your conflict of interest statement in the “Confidential to Editor” section, and submit your "Accept" recommendation.

Reviewer #3: All comments have been addressed

Reviewer #4: All comments have been addressed

2. Is the manuscript technically sound, and do the data support the conclusions?

Reviewer #3: Yes

Reviewer #4: Partly

3. Has the statistical analysis been performed appropriately and rigorously? 

Reviewer #3: N/A

Reviewer #4: Yes

4. Have the authors made all data underlying the findings in their manuscript fully available?

Reviewer #3: Yes

Reviewer #4: Yes

5. Is the manuscript presented in an intelligible fashion and written in standard English?

Reviewer #3: Yes

Reviewer #4: Yes

6. Review Comments to the Author

Reviewer #3: My overall impression is positive about this study. The article is well structured. The data provided can be easily studied as well as the results. The Minor points of this part of the healthcare system are accurately described and they are reflected through the results.

Reviewer #4: The authors have adopted most of the comments and suggestions. If possible, it is suggested to complement with the following recommendations:

Clarity and depth of quantitative analysis:

- It is recommended to explain in greater detail why specific statistical methods, such as the Mann-Whitney and Chi-square tests, were chosen. Additionally, a deeper discussion on the assumptions and limitations of these methods would strengthen the analysis.

- Providing a more detailed description of the coding process and thematic analysis, including examples of coded quotes and how they were grouped into themes, would improve the transparency and replicability of the qualitative analysis.

- Detailing the standard procedures used for training healthcare professionals in the implementation of ANC guidelines would offer a clearer view of how to improve adherence in other contexts.

- Proposing a specific system for the continuous evaluation of adherence to the guidelines, including key performance indicators and feedback methods for healthcare providers, would be beneficial.

- Including more concrete and practical suggestions to overcome the identified barriers, based on the study's findings, would enrich the discussion.

- Expanding the review of existing literature on adherence to ANC guidelines in other middle- and low-income countries would provide a broader context for the study's results.

- Discussing the study's limitations in more depth, especially regarding sample size and the generalizability of the findings, would provide a more balanced and critical view of the study.

- Identifying specific areas for future research, such as longitudinal studies that evaluate changes in adherence over time or research on the implementation of specific interventions to improve adherence, would be beneficial.

7. PLOS authors have the option to publish the peer review history of their article (what does this mean?). If published, this will include your full peer review and any attached files.

Reviewer #3: No

Reviewer #4: No

---

## [Author Response · Author response to Decision Letter 2]

9 Aug 2024

Journal Requirements:

Author response:

Thank you for reviewing our work in detail. Previously, we used some unpublished articles and regulations from the Ministry of Health of the Republic of Indonesia as references for regulations and guidelines for implementing ANC in Indonesia. We also referred to regional government regulations for adjustments to ANC guidance. However, for clarity and compliance with writing rules, we have replaced some unpublished and retracted references. Nonetheless, we have retained references to guidelines for implementing antenatal care as a standard measure for determining health workers' adherence to ANC guidelines. The specific changes we have made are as follows:

1. We regret the oversight in the previous reference numbers. Reference numbers 15, 20, and 41-44 have been revised to ensure compliance with writing guidelines. These references contain vital data on health profile reports and health performance achievements at both the national and regional levels in Indonesia.

2. Furthermore, we acknowledge and apologize for the oversight in reference numbers 32, 33, 34, 56, and 57. These reference numbers are pertinent to a government policy being tested at the Community Health Center and, as a result, must remain unchanged.

3. We have taken steps to rectify the duplication error in reference number 22.

4. Reference number 35 has been updated, and reference number 63 has been revised to conform to the rules of writing thesis references. The information obtained from reference number 63 serves as crucial background information for the findings in this research.

5. We also have added the abbreviations of some acronyms for clarity (PONED and PONEK)

Comments to the Author

2. Is the manuscript technically sound, and do the data support the conclusions?

Reviewer #3: Yes

Reviewer #4: Partly

Reviewer #4: The authors have adopted most of the comments and suggestions. If possible, it is suggested to complement with the following recommendations:

Clarity and depth of quantitative analysis:

- It is recommended to explain in greater detail why specific statistical methods, such as the Mann-Whitney and Chi-square tests, were chosen. Additionally, a deeper discussion on the assumptions and limitations of these methods would strengthen the analysis.

Author response:

Thank you for your suggestion. We have added the rationale for the chosen statistical methods on (page 12 line 251-252).

The selection of hypothesis tests is based on a variable scale, categorized into 2 to 3 groups with commonly used demographic characteristics.

Thank you for your detail review. We do agree to add explanation regarding the limitation of Mann-Whitney and Chi square test in the discussion section. We have add the limitation of statistical methods after the previous sentence below (page 23 line 472-480): 

The sentence added: 

The results of this study are affected by the number of samples and the limitations of the chi-square statistical test that was used. This test is very sensitive to sample size and is more suitable for studies with a large number of samples. In this particular study, multiple sample categories have been combined to ensure a larger sample size, but this may not completely overcome the limitations of the chi-square test. As a result, a larger sample size is needed in the next study in order to effectively compare the services offered by the two types of Community Health Centers. This will ensure that the expected frequency in each group is more than 5, allowing for the use of the chi-square test.

- Providing a more detailed description of the coding process and thematic analysis, including examples of coded quotes and how they were grouped into themes, would improve the transparency and replicability of the qualitative analysis.

Thank you for your comprehensive review. We have revised the text to provide more clarity by including details of the coding and analysis as follows: (page 13 line 289-301)

They coded significant statements manually, labelled them with the participants' keywords and phrases, and cross-checked and discussed all the codes. Analysis was adaptive, integrating thematic areas that researchers had generated with input process and output models as components of the health system. The consensus was reached through regular analysis discussions, and we had differences of opinion to including the code limited health provider as an imbalance resource or role responsibility, and we consulted this difference to the research supervisor (AU). Finally, we note that a limited number of health care providers are included in the theme of Imbalance resource. Although it meets the standards of health centers in urban areas, it may not be sufficient for the number of residents in the health center's work area.

The flowchart for coding acquisition is shown in figure 1:

 - Detailing the standard procedures used for training healthcare professionals in the implementation of ANC guidelines would offer a clearer view of how to improve adherence in other contexts.

Thank you for your detail review, to clarify the recommendation of this study, we have added explanation of specific training needed in page 29 line 626-633

It is essential to provide collaborative training for midwives, dentists, and general practitioners on integrated antenatal care procedures and pregnancy complication screening. This will ensure that multi-professional services can be effectively carried out in both types of health centers. When scheduling the training, it's important to consider the service hours at the health center in order to maximize the participation of invited health workers, especially at non-Basic Emergency Obstetric and Newborn Care (BEmONC) health centers, without disrupting service hours. 

And move the sentence: The blood type examination was identified as an inefficient laboratory examination. To the appropriate paragraph.

We also have identified unique data findings in table 3 and included corresponding sentences in the discussion section to explain the findings. (page 15 line 316-324)

Table 3 reveals that there was a lack of adherence to antenatal care services in one pregnant woman under the age of 20 and 1 pregnant woman (1%) with more than 3 previous pregnancies. It is important to further investigate whether there are barriers such as shame or fear preventing pregnant women from undergoing antenatal care examinations as per national guidelines. Unfortunately, the researchers were unable to interview the patient, which hindered the exploration of the patient's experience with the service. Future research should aim to include a more balanced representation from the age group under 20 and women with more than 3 previous pregnancies in order to gain a comprehensive understanding of antenatal care services within this demographic.

- Proposing a specific system for the continuous evaluation of adherence to the guidelines, including key performance indicators and feedback methods for healthcare providers, would be beneficial.

Author response:

Thank you for the very detailed review. We have added sentences to explain the evaluation as advised. (page 26 line 566-571)

Pharmacy officers do not fully understand how to use various applications as tools to carry out stocktaking more effectively and efficiently, so they still rely on manual methods. Despite the mandatory paperless policy and electronic medical records introduced by the head of the Semarang City Health Service, it is essential to conduct evaluations. This includes assessing the readiness of the officers, providing regular training, and developing the system based on feedback and suggestions from those implementing it. 

- Including more concrete and practical suggestions to overcome the identified barriers, based on the study's findings, would enrich the discussion.

Author response:

Thank you for the very detailed review. We have added sentences to explain the evaluation as advised. Page 27 line 591-597)

To implement the policy of regionalizing health services, it is important to establish clear guidelines for cooperation between the city leadership of Semarang and the leadership of BPJS-K. These guidelines should ensure that the community can access services according to their needs, without being restricted by regional boundaries. It's crucial to avoid overburdening specific health facilities while meeting the public's demand for services. Additionally, equitable development of health facilities throughout Semarang is necessary to ensure that high-quality healthcare is accessible to the entire community. 

We also move the sentece 

In addition, the ANC form does not mention general physical examination, which causes the health workers to forget to write down the results of medical check-ups in the medical record. (Page 24 line 489-492)

To the appropriate paragraph (page 24 line 502-504) and we have added the relevant recomendation as follow: (page 25 line 522-525)

It is essential to include a checklist of physical examination results in the patient's medical record. This checklist should contain the outcomes of all integrated antenatal care examination procedures to minimize unrecorded examinations by health workers. 

- Expanding the review of existing literature on adherence to ANC guidelines in other middle- and low-income countries would provide a broader context for the study's results.

Author response:

Thank you for the very detailed review. We have added sentences to explain the evaluation as advised.

Page 27 line 579-584

The issue of regionalization and uneven distribution of healthcare workers is also evident in several low- and middle-income countries (LMICs). For example, in rural districts of Tanzania, there are often more than 30–50 new first antenatal care (ANC) visits per month. Many of these women come from outside the local area, traveling from neighboring districts and surrounding wards that have fewer health facilities. 

Page 25 line 536-541

Several low- and middle-income countries (LMICs) have faced serious incidents hindering the implementation of antenatal care (ANC). For instance, in Tanzania, some health facilities have issues such as moldy walls, damaged roofs and windows, and bird nests inside the facility. Additionally, many facilities lack access to water and electricity, frequently run out of essential medicines, and do not have maternal and child health cards or books to record ANC visit history and examination results.

Page 25 line 555-557

Studies in Rwanda also indicate the need to improve room facilities, availability of competent health providers, and interprofessional collaboration to enhance the quality of integrated ANC implementation.

- Discussing the study's limitations in more depth, especially regarding sample size and the generalizability of the findings, would provide a more balanced and critical view of the study.

Author response:

Thank you for the very detailed review. We have added sentences to explain the evaluation as advised. (page 28 line 607-608)

The research findings may not be generalizable due to the sensitivity of the statistical tests to sample size. 

- Identifying specific areas for future research, such as longitudinal studies that evaluate changes in adherence over time or research on the implementation of specific interventions to improve adherence, would be beneficial.

Author response:

Thank you for the very detailed review. We have added sentences to explain the evaluation as advised. (page 29 line 633-639)

 It is also important to conduct longitudinal research to assess officer compliance after implementing various recommendations from this study. This includes ensuring infrastructure fulfillment, improving the format of ANC patient medical records, providing training on procedures for implementing collaborative integrated antenatal care, and aligning health service regionalization policies. This will enable continuous improvement in integrated antenatal care services to meet the high-quality health needs of pregnant women later on.

---

## [Editor Report · Decision Letter 3]

13 Aug 2024

A mixed-method analysis of provider adherence to integrated antenatal care guideline in BEmONC and Non BEmONC primary health center: An Indonesian case

PONE-D-23-12582R3

Dear Dr. Yuliyanti,

We’re pleased to inform you that your manuscript has been judged scientifically suitable for publication and will be formally accepted for publication once it meets all outstanding technical requirements.

Kind regards,

Gilbert Sterling Octavius

Academic Editor

PLOS ONE
---

## [Editor Report · Acceptance letter]

16 Aug 2024

PONE-D-23-12582R3 

PLOS ONE

Dear Dr. Yuliyanti, 

I'm pleased to inform you that your manuscript has been deemed suitable for publication in PLOS ONE. Congratulations! Your manuscript is now being handed over to our production team.

Kind regards, 

on behalf of

Dr. Gilbert Sterling Octavius 

Academic Editor

PLOS ONE